# Towards Transferable Adversarial Perturbations with Minimum Norm

**Fangcheng Liu**[1]   **Chao Zhang**[1]   **Hongyang Zhang**[2]

## Abstract

Transfer-based adversarial example is one of the most important classes of black-box attacks. Prior work in this direction often requires a fixed but large perturbation radius to reach a good transfer success rate. In this work, we propose a *geometry-aware framework* to generate transferable adversarial perturbation with minimum norm for each input. Analogous to model selection in statistical machine learning, we leverage a validation model to select the optimal perturbation budget for each image. Extensive experiments verify the effectiveness of our framework on improving image quality of the crafted adversarial examples. The methodology is the foundation of our entry to the CVPR'21 Security AI Challenger: Unrestricted Adversarial Attacks on ImageNet, in which we ranked 1st place out of 1,559 teams and surpassed the runner-up submissions by 4.59% and 23.91% in terms of final score and average image quality level, respectively.

## 1. Introduction

Though deep neural networks have exhibited impressive performance in various fields (He et al., 2016; Dosovitskiy et al., 2021), they are vulnerable to adversarial examples (Szegedy et al., 2014; Goodfellow et al., 2015; Bhojanapalli et al., 2021; Shao et al., 2021), where test inputs that have been modified slightly strategically cause misclassification. Adversarial examples have posed serious threats to various security-critical applications, such as autonomous driving (Bojarski et al., 2016), face recognition (Parkhi et al., 2015; Zhong & Deng, 2020), malware classification (Pascanu et al., 2015), etc. Most positive results on adversarial attacks have focused on the white-box settings (Athalye et al., 2018; Tramer et al., 2020), where the attacker can get full access to the defense models. However, the problem

[1]Peking University [2]University of Waterloo & Vector Institute. Correspondence to: Chao Zhang <c.zhang@pku.edu.cn>, Hongyang Zhang <hongyang.zhang@uwaterloo.ca>.

*Accepted by the ICML 2021 workshop on A Blessing in Disguise: The Prospects and Perils of Adversarial Machine Learning.* Copyright 2021 by the author(s).

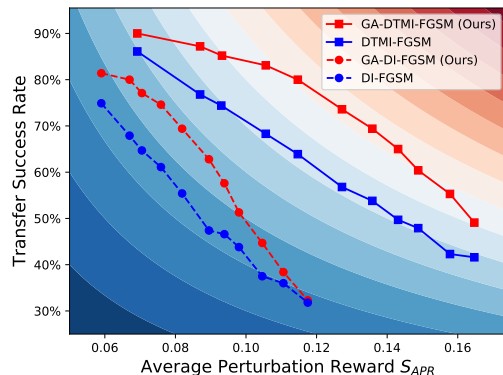

*Figure 1.* Contour of average total score $S_{total}$ (see Eq (2), the upper right corner represents a better result), which is defined as the product of transfer success rate (y-axis) and average perturbation reward $S_{APR}$ (x-axis). Fixing transfer success rate as 80%, our approach GA-DTMI-FGSM surpasses the baseline DTMI-FGSM (see Eq (1)) by up to 43.35% in terms of average perturbation reward, which benefits from the adaptive choice of perturbation budgets w.r.t. distinct images in our geometry-aware framework.

becomes more challenging when it comes to the black-box setting, where the attacker has no information about the model architecture, hyper-parameters, and even the model outputs. In this setting, adversarial examples are typically generated via *transfer-based* methods (Szegedy et al., 2014; Papernot et al., 2016; 2017), e.g., attacking an ensemble of accessible source models and hoping that the same perturbations are able to fool the unknown target (test) model (Liu et al., 2017; Tramèr et al., 2017).

Despite a large amount of work on transfer-based attacks, many fundamental questions remain unresolved. For example, existing transfer-based attacks (Dong et al., 2018; Xie et al., 2019; Dong et al., 2019) that search for adversarial examples in a fixed-radius ball often require high perturbation budget to reach a good transfer success rate. However, such perturbations might be perceptible to humans (see Figure 2). On the other hand, minimum-norm attacks (Carlini & Wagner, 2017) that fool the source model with minimum perturbation suffer from weak transferability to the target model. This is in part due to the difference between the decision boundaries of the source and target models. Given the difficulty of trading transferability off against perturbation budget, one of the long-standing questions is generating minimum-norm adversarial perturbations that can transfer

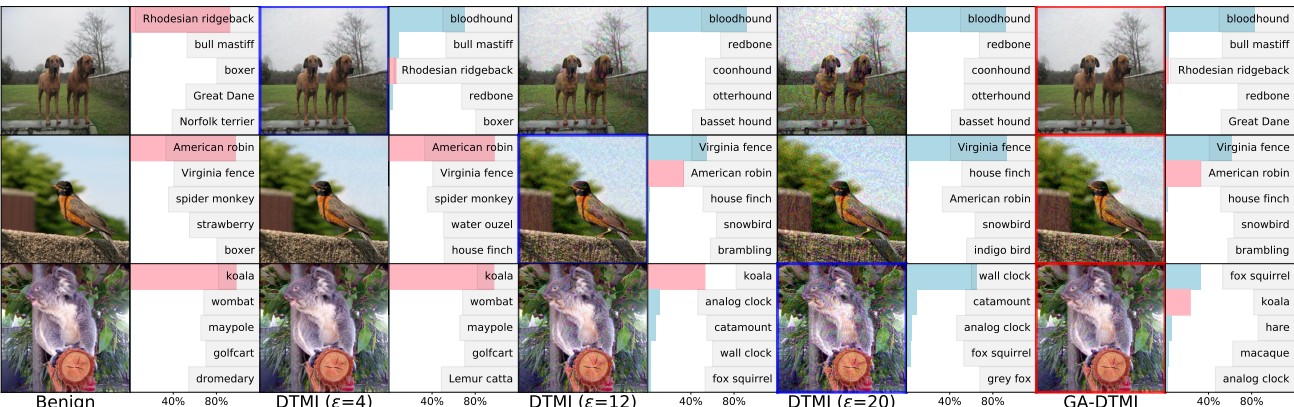

*Figure 2.* Comparison between our method GA-DTMI-FGSM and the baseline DTMI-FGSM (see Eq (1)) using various $\ell_\infty$ perturbation radii. In the even columns, we present the top-5 confidence bars of the target (test) model for the images in the left. The ground-truth label is marked by pink and other labels are marked by blue. In each row, the misclassified adversarial example with minimum perturbation norm under DTMI-FGSM attack is highlighted by a blue bounding box. This indicates that the perturbation budgets required for distinct images are different. Note that the "human-imperceptible" constraint is violated when the perturbation radius is too large. However, our method GA-DTMI-FGSM (highlighted by red bounding boxes) generates transferable adversarial examples with lower budget.

well across various networks.

### 1.1. Our methodology and results

In this work, we propose a novel geometry-aware framework, into which existing fixed-radius methods can be integrated to generate transferable adversarial perturbation with minimum norm. Our intuition is that the smallest perturbation budgets w.r.t. distinct images should be different (see Figure 2) and should depend on their geometrical distances to the decision boundary of the target model. Unfortunately, finding transferable minimum-norm adversarial perturbations is an intractable optimization problem (see Eq (3)). We approximately solve this problem by discretizing the continuous space of perturbation radius into a finite set and choosing the minimum perturbation budget that is able to fool the test model. The main challenge here is to evaluate whether a given perturbation transfers well to the unknown target model (Cheng et al., 2019b; Katzir & Elovici, 2021).

To overcome this challenge, we split all accessible white-box source models into training and validation sets, where adversarial perturbations are crafted only on the training set. We use the validation set to select the smallest perturbation radius for each input that suffices to fool the validation model with a certain confidence level through an early-stopping mechanism. When the training (or validation) set consists of multiple models, we use their ensemble with equal weights (Liu et al., 2017). Experimentally, our method yields significant performance boost on the trade-off between average perturbation reward (see Eq (2)) and transfer success rate. As shown in Figure 1, the transfer success rate of our method GA-DTMI-FGSM surpasses the baseline DTMI-FGSM (see Eq (1)) by up to 16% in absolute value under the same average perturbation reward.

### 1.2. Summary of our contributions

- We propose a geometry-aware framework, where existing fixed-radius methods can be integrated, to generate transferable adversarial perturbation with minimum-norm for each input. To the best of our knowledge, we are the first to explore the method of using adaptive perturbation radius in the transfer-based attacks.

- Our method improves the image quality of the crafted adversarial examples by a large margin without the decrease of transfer success rate. By applying our method to the CVPR'21 Security AI Challenger: Unrestricted Adversarial Attacks on ImageNet, we ranked 1st place out of 1,559 teams and surpassed the runner-up submissions by 4.59% and 23.91% in terms of final score and average image quality level, respectively.

## 2. Preliminaries

**Notation.** A deep neural network classifier can be described as a function $f(\boldsymbol{x}; \boldsymbol{\theta}) : \mathcal{X} \to \mathbb{R}^C$, parameterized by weights $\boldsymbol{\theta}$, which maps a vector $\boldsymbol{x} \in \mathcal{X}$ to its output logits. Given an input $\boldsymbol{x}$ of class $y \in \{1, 2, \cdots, C\}$, the predicted label of $f(\boldsymbol{x}; \boldsymbol{\theta})$ is $\hat{f}(\boldsymbol{x}) := \arg\max_j f_j(\boldsymbol{x}; \boldsymbol{\theta})$, where $f_j(\boldsymbol{x}; \boldsymbol{\theta})$ represents the $j$-th entry of $f(\boldsymbol{x}; \boldsymbol{\theta})$. We use $L(f(\boldsymbol{x}; \boldsymbol{\theta}), y)$ to represent the cross-entropy loss and denote the $\varepsilon$-neighborhood of $\boldsymbol{x}$ by $\mathbb{B}(\boldsymbol{x}, \varepsilon) := \{\boldsymbol{x}' \in \mathcal{X} : \|\boldsymbol{x}' - \boldsymbol{x}\| \leq \varepsilon\}$. We denote the black-box test model by $g$, and split the set of white-box source models into the set of training models $f$ and the set of validation models $h$.

### 2.1. Transfer-based black-box attacks

Adversarial perturbations can transfer well across different networks that are trained on similar tasks (Papernot et al.,

2016; 2017): adversarial examples that are generated on training model $f$ may also be misclassified by test model $g$. This intriguing property can be leveraged to perform transfer-based attacks. Existing transfer-based attacks typically search for adversarial examples in a fixed-radius ball.

**Fixed-radius $\ell_\infty$ perturbations.** Various methods were proposed to boost transferability of adversarial examples, such as input Diversity Iterative Fast Gradient Sign Method (DI-FGSM) (Xie et al., 2019), Momentum-based Iterative (MI-FGSM) method (Dong et al., 2018) and Translation-invariant Iterative (TI-FGSM) method (Dong et al., 2019). We formulate a strong baseline DTMI-FGSM by combining all these techniques, i.e.,

$$
\begin{aligned}
\boldsymbol{m}_{t+1} &= \mu \cdot \boldsymbol{m}_t + \frac{\mathbf{W} * \nabla_{\boldsymbol{x}_t} L\left(f\left(T(\boldsymbol{x}_t, p); \boldsymbol{\theta}\right), y\right)}{\|\mathbf{W} * \nabla_{\boldsymbol{x}_t} L\left(f\left(T(\boldsymbol{x}_t, p); \boldsymbol{\theta}\right), y\right)\|_1}, \\
\boldsymbol{x}_{t+1} &= \Pi_{\mathbb{B}(\boldsymbol{x}, \varepsilon)}\left(\boldsymbol{x}_t + \alpha \cdot \text{sign}(\boldsymbol{m}_{t+1})\right),
\end{aligned}
\tag{1}
$$

where $\boldsymbol{m}_0 = \boldsymbol{0}$, $\mathbf{W}$ is a pre-defined kernel with a convolution operation $*$, $\alpha$ is the step size, $\Pi$ is the projection operator, and $\mu$ is the decay factor for the momentum term. $T(\boldsymbol{x}_t, p)$ represents the input transformation on $\boldsymbol{x}_t$ with probability $p$. When $\mu = 0$, DTMI-FGSM attack degenerates to the DTI-FGSM attack.

## 2.2. Evaluation metric

It is important to keep high transfer success rate while maintaining "human-imperceptible" constraint and good perceptual quality such that the crafted adversarial examples are not easily detected (Grosse et al., 2017) by human inspectors. Although the precise mathematical quantization of human perception is difficult to obtain (Wang et al., 2004; Laidlaw et al., 2021), typically smaller perturbation radius enjoys better image quality (see Figure 2). Consider a dataset $\hat{S} = \{(\boldsymbol{x_i}, y_i)\}_{i=1}^n$ and the corresponding adversarial examples $\hat{S}_{\text{adv}} = \{(\boldsymbol{x'_i}, y_i)\}_{i=1}^n$ that are crafted on the training model $f$. Let $n_0 = \sum_{i=1}^n \mathbb{1}\{\hat{g}(\boldsymbol{x'_i}) \neq y_i\}$ be the number of misclassified adversarial examples on test model $g$. We define the average total score as follows:

$$
\begin{aligned}
S_{total} &= \frac{1}{n} \sum_{i=1}^n \mathbb{1}\{\hat{g}(\boldsymbol{x'_i}) \neq y_i\} \cdot \mathcal{F}_{\text{reward}}\left(\|\boldsymbol{x'_i} - \boldsymbol{x_i}\|\right) \\
&= \frac{n_0}{n} \cdot \frac{1}{n_0} \sum_{i=1}^n \mathbb{1}\{\hat{g}(\boldsymbol{x'_i}) \neq y_i\} \cdot \mathcal{F}_{\text{reward}}\left(\|\boldsymbol{x'_i} - \boldsymbol{x_i}\|\right) \\
&\stackrel{\text{def}}{=} \frac{n_0}{n} \cdot S_{APR},
\end{aligned}
\tag{2}
$$

where $S_{APR}$ is the Average Perturbation Reward of adversarial examples that are misclassified by test model $g$ and the reward function $\mathcal{F}_{\text{reward}}$ is a decreasing function w.r.t. the perturbation raidus $\|\boldsymbol{x'_i} - \boldsymbol{x_i}\|$.

## 3. Methodology

**Motivation.** Eq (2) factorizes the average total score as the product of transfer success rate and average perturbation reward, which motivates us to find the smallest perturbation $\boldsymbol{\delta}^*$ such that $\boldsymbol{x} + \boldsymbol{\delta}^*$ is misclassified by the test model $g$,[1] i.e.,

$$
\boldsymbol{\delta}^* = \arg\min_{\boldsymbol{\delta}} \|\boldsymbol{\delta}\|, \text{ s.t. } \hat{g}(\boldsymbol{x} + \boldsymbol{\delta}) \neq y.
\tag{3}
$$

However, direct optimization of problem (3) is intractable, in part due to the lack of information about test model $g$. We approximately solve this problem by discretizing the continuous space of perturbation radius into a discrete set and choosing the minimum perturbation budget such that the attack is able to fool the test model $g$. However, the challenge is that it is typically difficult to decide whether a given perturbation radius can also fool the test model (Cheng et al., 2019b; Katzir & Elovici, 2021). This problem is also known as model selection problem, and a classic approach to tackle this problem is to use a validation model $h$ to select the right perturbation radius. More specifically, we split all accessible source models into training model set and validation model set. With validation model $h$, we are able to generate transferable adversarial examples with smaller perturbation budgets.

### 3.1. Geometry-aware framework for generating transferable minimum-norm perturbations

Existing fixed-radius transfer-based methods typically ignore the *local geometry* property, i.e., the geometrical distances of distinct images to the decision boundary of test model $g$ are different. To approximately solve problem (3), we first divide the attack in the ball $\mathbb{B}(\boldsymbol{x}, \varepsilon)$ into $K$ sub-procedures. In the $k$-th sub-procedure, we re-run a fixed-radius attack algorithm $\mathcal{A}$ such as DTMI-FGSM (see Eq (1)), under the perturbation budget

$$
\varepsilon_k = \frac{k}{K} \times \varepsilon, \ k = 1, 2, \cdots, K.
$$

We start each sub-procedure from the solution of the last sub-procedure to accelerate the convergence. To obtain a minimum-norm solution, we perform an early-stopping mechanism at the end of each sub-procedure if the probability of the true class on the validation model $h$ is smaller than a certain threshold $\eta$, i.e.,

$$
\mathbb{P}\left(\hat{h}(\boldsymbol{x}) = y\right) = \frac{\exp\left(h_y(\boldsymbol{x}; \boldsymbol{\theta})\right)}{\sum_j \exp\left(h_j(\boldsymbol{x}; \boldsymbol{\theta})\right)} < \eta.
\tag{4}
$$

Our geometry-aware framework is summarized in Algorithm 1 and conceptually described in Figure 3.

---

[1] In contrast to existing white-box minimum-norm attack (Carlini & Wagner, 2017), $g$ is a unknown model here.

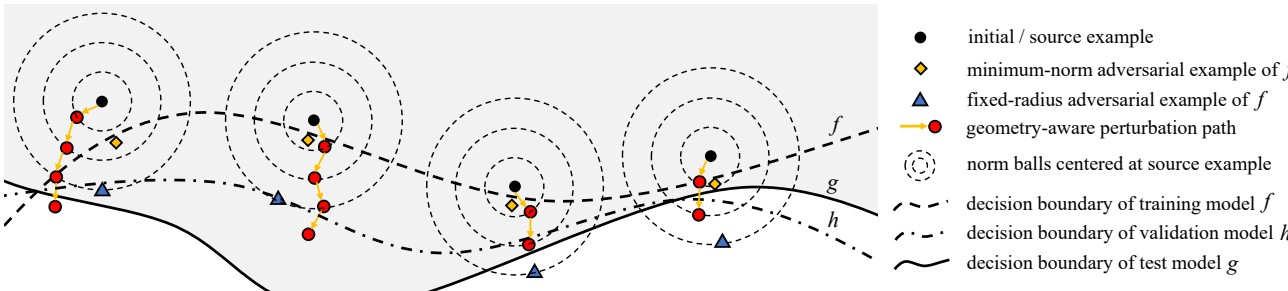

*Figure 3.* Existing fixed-radius methods typically overlook the the importance of geometrical distances from inputs to the decision boundary of test model $g$. In contrast, our geometry-aware framework aims to find geometry-aware minimal-norm perturbation via a validation model $h$. The goal is to prevent an attack algorithm overfitting $f$ by forcing the solution to cross the decision boundary of $h$ with a certain margin. Our geometry-aware framework consists of multiple sub-procedures with adaptive perturbation budgets. In each sub-procedure, we start from the solution of last sub-procedure (the red solid points) and re-run the attack algorithm on the training model $f$. The procedure stops if the output probability of the true class on the validation model $h$ is smaller than a certain threshold $\eta$.

## 4. Experiments

In this section, we compare fixed-radius baselines with our geometry-aware framework. The methodology is the foundation of our entry to the CVPR'21 Security AI Challenger, in which we ranked 1st place out of 1,559 teams.

### 4.1. Improved trade-off between transfer success rate and average perturbation reward

We combine our Geometry-Aware framework with DI-FGSM, DTI-FGSM and DTMI-FGSM, named GA-DI-FGSM, GA-DTI-FGSM and GA-DTMI-FGSM, respectively. In our experiment, the training model $f$ and validation model $h$ are an ensemble of models $\{2, 3, 5\}$ and $\{1, 4, 6\}$ in Table 2, respectively. The test model is Inception-ResNet-v2[2]. As shown in Table 1, our approach yields significant performance boost on both the average perturbation reward $S_{APR}$ and the transfer success rate.

*Table 1.* Comparison of our geometry-aware framework with baselines. We report the results when both our approach and baselines achieve highest average total score $S_{APR}$.

| Method | Transfer Success Rate | $S_{APR}$ ($\uparrow$) | $S_{total}$ ($\uparrow$) |
|---|---|---|---|
| DI-FGSM | 61.1% | 0.0759 | 4.64% |
| GA-DI-FGSM (ours) | **69.4%** | **0.0819** | **5.68%** |
| DTI-FGSM | 57.3% | 0.1101 | 5.68% |
| GA-DTI-FGSM (ours) | **67.9%** | **0.1176** | **7.98%** |
| DTMI-FGSM | 63.9% | 0.1147 | 7.33% |
| GA-DTMI-FGSM (ours) | **69.4%** | **0.1358** | **9.42%** |

### 4.2. Case study: CVPR 2021 Competition on Unrestricted Adversarial Attacks on ImageNet

**Competition settings.** In the CVPR 2021 Competition on Unrestricted Adversarial Attacks on ImageNet, contestants were asked to submit adversarial examples without any access to the defense models. The dataset is a subset of ILSVRC 2012 validation set (Deng et al., 2009), which con-

[2]For more experimental setups, we refer to Appendix B.

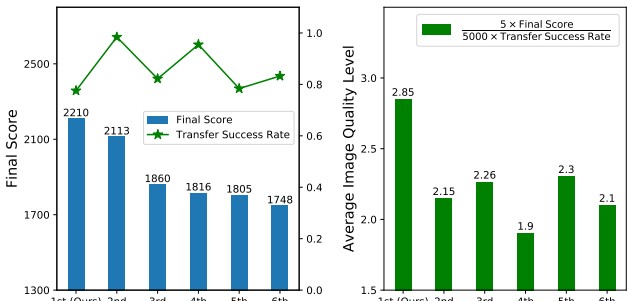

*Figure 4.* Top-6 results (out of 1,559 teams) in the CVPR'21 Security AI Challenger: Unrestricted Adversarial Attacks on ImageNet. The final scores were manually scored by multiple human referees.

sists of 5,000 images with 5 images per class. The final score of each submission was *manually* scored from two aspects: 1) image semantic and 2) quality. If the semantic of the submitted image changes, then $S_s = 0$, otherwise $S_s = 1$. The image quality $S_q$ (equivlent to our reward function $\mathcal{F}_{\text{reward}}$) was quantified with five levels $S_q \in \{1, 2, 3, 4, 5\}$ by multiple human referees. The final score is given by $\sum_i \mathbb{1}\{\hat{g}(\boldsymbol{x}'_i) \neq y_i\} \times S_s(\boldsymbol{x}'_i) \times \frac{S_q(\|\boldsymbol{x}'_i - \boldsymbol{x}_i\|)}{5}$.

**Competition results.** We apply our method GA-DTMI-FGSM to the competition, where our entry ranked 1st place out of 1,559 teams. We report the final score and average image quality level (equivlent to our average perturbation reward) in Figure 4. It shows that our method outperforms other approaches by a large margin. In particular, we surpass the runner-up submissions by 4.59% and 23.91% in terms of final score and average image quality level, respectively.

## 5. Conclusion

In this work, We propose a geometry-aware framework, where existing fixed-radius methods can be integrated to generate transferable adversarial perturbation with minimum-norm for each input.

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

## A. Related Works

**Generating $\ell_p$ adversarial examples.** The attacks for generating $\ell_p$ adversarial examples can be summarized into white-box attacks, *query-based* and *transfer-based* black-box attacks. In white-box threat model, attackers have full access to the defense model, such as gradients, parameters, logits, etc. Existing gradient-based white-box attacks either find adversarial examples in a fixed-radius ball (Goodfellow et al., 2015; Kurakin et al., 2019; Madry et al., 2018), or optimize the perturbation for each image independently to get a *minimum-norm* solution such as CW attack (Carlini & Wagner, 2017), DeepFool (Moosavi-Dezfooli et al., 2016), Fast Adaptive Boundary (FAB) attack (Croce & Hein, 2020) and Fast Minimum-Norm (FMN) attack (Pintor et al., 2021). However, white-box assumption usually does not hold in real-world scenarios; In query-based black-box threat model, attackers utilize the output logits (score-based attacks (Chen et al., 2017; Andriushchenko et al., 2020)) to generate adversarial examples, or only use the output label of the target model (decision-based attacks (Brendel et al., 2018; Cheng et al., 2019a)). But all these query-based attacks typically suffer from high query complexity, making it easy to be detected (Willmott et al., 2021); In transfer-based black-box threat model, attackers have no information about the defense model. Dong et al. (Dong et al., 2018) boosted transferability by integrating momentum into iterative gradient-based methods. Liu et al. (Liu et al., 2017) found that attacking a group of substituted source models simultaneously can improve transferability. Besides, transferability benefits from input transformations such as Diversity Iterative Fast Gradient Sign Method (Xie et al., 2019) and Translation-invariant Iterative Method (Dong et al., 2019). Recent work (Wang et al., 2021) shows that momentum-based method can be further enhanced by using gradients of more data points.

**Adversarial defenses.** There have been long-standing arms races between defenders and attackers: while many defense methods claimed non-trivial robustness, majority of them were later broken by adaptive attacks (Athalye et al., 2018; Tramer et al., 2020). Adversarial training (Goodfellow et al., 2015) is one of the most promising defense methods. There are various variants of adversarial training framework, e.g., ensemble adversarial training (Liu et al., 2017) for transfer-based attack, PGD-based adversarial training (Madry et al., 2018), and TRADES (Zhang et al., 2019) with a new robust loss based on the trade-off between robustness and accuracy. Geometry-aware instance-reweighted adversarial training (Zhang et al., 2021) shares similar insights with us that the importance of distinct inputs in adversarial training

should be different, however, is proven falling into gradient masking (Hitaj et al., 2021; Carlini & Wagner, 2016). Schmidt et al. (Schmidt et al., 2018) proved that adversarial robust generalization requires more data in a Gaussian model. Recent work shows that adding more training data can improve adversarial robustness, either unlabeled (Carmon et al., 2019) or synthetic data (Rebuffi et al., 2021; Sehwag et al., 2021).

**Beyond $\ell_p$ norms.** The $\ell_p$ norm distance is not an ideal perceptual similarity metric (Johnson et al., 2016; Isola et al., 2017), which oversimplifies the diversity of real-world perturbations. Kang et al. (Kang et al., 2019) found that defenses trained on $\ell_p$ bounded perturbation are not robust against new types of unseen attacks. Adversarial training against multiple $\ell_p$ attacks (Tramer & Boneh, 2019) solved this issue partially, however, at the cost of efficiency and robustness against single $\ell_p$ attack. Recent works (Laidlaw et al., 2021) integrate adversarial training with Learned Perceptual Image Patch Similarity (LPIPS), aiming to improving robustness against perturbations that were unseen during training. Kireev et al. (Kireev et al., 2021) introduced an efficient relaxation of perceptual adversarial training based on layer-wise adversarial perturbations. Instead of constructing adversarial examples on perceptual distances, Wong et al. (Wong & Kolter, 2021) used a conditional generator to define the perturbation set over a constrained region of the latent space. *Unrestricted adversarial examples* has received significant attention in recent years (Brown et al., 2018; Song et al., 2018; Bhattad et al., 2020; Engstrom et al., 2019; Zhao et al., 2020). However, constructing transferable unrestricted adversarial examples is still an open problem.

## B. Experiments

### B.1. Experimental setup

**Datasets & Networks.** Attacking on images that are already misclassified is trivial. Similar to (Xie et al., 2019), we randomly select 1,000 images from ILSVRC 2012 validation set (Deng et al., 2009), which are almost correctly classified by all the considered models. All these images are resized to $229 \times 229 \times 3$ beforehand. We consider eight normally trained models and two ensemble adversarially trained models (Tramèr et al., 2018). The weights of all these models are publicly available at Wightman (2019). More details about the networks are summarised in Table 2. The transferability between all considered models is summarised in Figure 5. The generated adversarial examples are much easier to transfer from vision transformers to CNNs, which is consist with the empirical observation in Shao et al. (2021). Surprisingly, the robustness of naturally trained vision transformers under transfer attack is even on par with two ensemble adversarially trained CNNs.

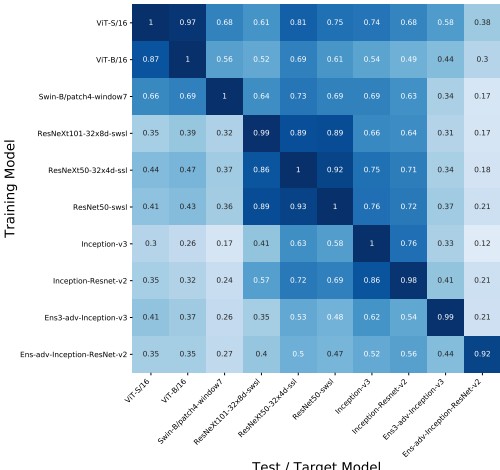

*Figure 5.* Transferability comparison between different networks under DTMI-FGSM attack. The rows stand for the source models and the columns stand for the target models. The crafted adversarial examples are much easier to transfer from vision transformers to CNNs. Surprisingly, the robustness of naturally trained vision transformers under transfer attack is even on par with two ensemble adversarially trained CNNs. Besides, adversarial examples transfer well between models with similar architectures.

**Implementation details.** For our geometry-aware framework, we set $K = 5$ and the maximum perturbation size $\varepsilon = 20$. The reward function is set to $\mathcal{F}_{\text{reward}}(x) = 1/x$. The optimal threshold $\eta^*$ was chosen from a finite set ranging from 0.001 to 0.9. For DTMI-FGSM, we set the step size $\alpha = \frac{1.25 \times \varepsilon}{T}$ with $T = 10$. For fair comparison, we set the iteration number $N = (0.125\varepsilon + 0.5) \times T$ for baselines to keep the same total perturbation budget as our geometry-aware framework. For the momentum term, we set the decay factor $\mu = 1$ as in Dong et al. (2018). For DI-FGSM (Xie et al., 2019), we set the transformation probability to $p = 0.7$ and the input $x$ is first randomly resized to an $r \times r \times 3$ image with $r \in [(1 - \gamma)s, (1 + \gamma)s]$, and then padded to size $(1 + \gamma)s \times (1 + \gamma)s \times 3$. The transformed input is then resized to $s \times s \times 3$ for different input size $s$ of various models, i.e., 224, 299 or 384. We set $\gamma = 0.1$ as default. For TI-FGSM (Dong et al., 2019), we use Gaussian kernel with kernel size equals to $5 \times 5$.

### B.2. Ablation Studies

**The optimal $\eta$ depends on the combination of $f, h, g$** It is natural to ask that how should we choose the training model $f$ and validation model $h$. As illustrated in Figure 3, the effectiveness of our geometry-aware framework relies on relationship between the decision boundaries of training model $f$, validation model $h$ and test model $g$. Thus we compared various kinds of partition of training models, validation models and test model in Figure 7, from which we can conclude the following observations:

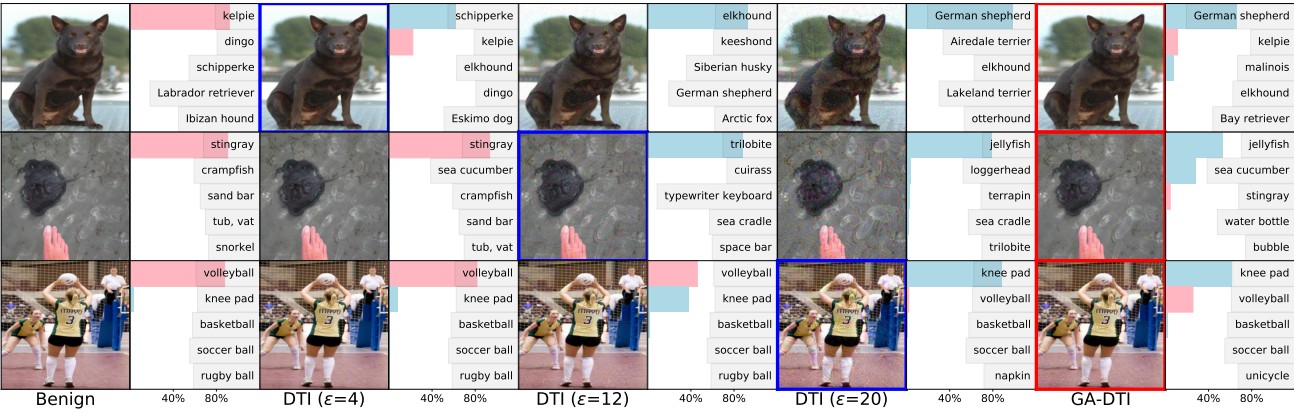

*Figure 6.* Comparison between our method GA-DTI-FGSM and the baseline DTI-FGSM using various $\ell_\infty$ perturbation radii. In the even columns, we present the top-5 confidence bars of the target (test) model for the images in the left. The ground-truth label is marked by pink and other labels are marked by blue. In each row, the misclassified adversarial example with minimum perturbation norm under DTI-FGSM attack is highlighted by a blue bounding box. Our method GA-DTI-FGSM (highlighted by red bounding boxes) generates transferable adversarial examples with lower budget.

*Table 2.* An overview of all considered networks. Top-1 represents the accuracy on the ILSVRC 2012 validation set (Deng et al., 2009).

| Training Method | Index | Model Name | Top-1 | Index | Model Name | Top-1 |
|---|---|---|---|---|---|---|
| Normal | 0 | ViT-S/16 | 76.01% | 1 | ViT-B/16 | 81.08% |
| | 2 | Swin-B/patch4-window7 | **84.23%** | 3 | ResNeXt101-32x8d-swsl | 83.62% |
| | 4 | ResNeXt50-32x4d-ssl | 78.90% | 5 | ResNet50-swsl | 79.97% |
| | 6 | Inception-v3 | 76.94% | 7 | Inception-ResNet-v2 | 79.85% |
| Ensemble | 8 | Ens3-adv-Inception-v3 | 76.49% | 9 | Ens-adv-Inception-ResNet-v2 | 78.98% |

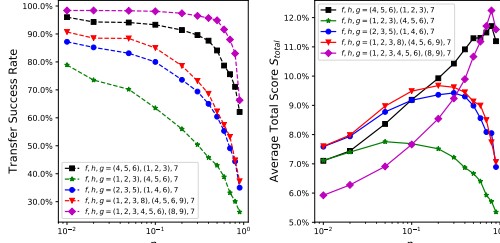

*Figure 7.* Comparison between various kinds of partition of training models, validation models and test model. The index in the legend corresponds to the model index in Table 2. As the threshold $\eta$ for early-stopping (see Eq (4)) increases, the generated adversarial examples have higher confidence (probability of the true class) on the validation model $h$, thus leading to a lower transfer success rate (**left**). Besides, the optimal $\eta$ that yields maximum average total score is dependent on the partition of the models (**right**).

- The role of training model $f$ and validation model $h$ is not *symmetrical*. The transfer success rate changed a lot when we exchanged the role of models $\{4, 5, 6\}$ and $\{1, 2, 3\}$ for training or validation.

- Ensembling more models yields better performance. When additional models were added into the training and validation sets, the average total score achieves significantly improvement.

**Algorithm 1** Geometry-Aware Framework for Transferable Minimum-Norm Perturbations

**Require:**
    Benign input $x$ with label $y$; training models $f$; validation model $h$; number of sub-procedures $K$; maximum perturbation size $\varepsilon$ and threshold $\eta$; attack algorithm $\mathcal{A}$

**Ensure:**
    Transfer-based adversarial example $x'$;

1:  $x_0 = x$;
2:  **for** $k = 1, 2, \cdots, K$ **do**
3:     $x_k = \mathcal{A}(x, x_{k-1}, f, \frac{k\varepsilon}{K})$;
4:     $\text{conf} \leftarrow \frac{\exp(h_y(x_k; \theta))}{\sum_j \exp(h_j(x_k; \theta))}$;
5:     **if** $\text{conf} < \eta$ **then**
6:         **Return** $x_k$;
7:     **end if**
8:  **end for**
9:  **Return** $x_K$;