# OpenReview forum: "Towards Transferable Adversarial Perturbations with Minimum Norm"
_ICML.cc/2021/Workshop/AML — ICML 2021 Workshop AML Poster_

### Official Review · Reviewer_SCAq · 2021-06-19
**The proposed geometry-aware framework is effective in generating adversarial examples with minimum perturbation norm.**

**Rating:** Accept
**Confidence:** 4

**Review:**

Pros:
1. The writing of the paper is good.
2. The experiment results are sufficient.
3. The transfer from unrestricted adversarial attack to a minimization of perturbation norm is clever.
Cons:
Some comparative experiments with different maximum perturbation settings should be done, as the 20/255 perturbation may be too large to some of the data samples.

---

### Decision · Program_Chairs · 2021-06-21

**Decision:**

Accept (Poster)

**Comment:**

A good paper to generate transferable adversarial examples with minimum norm. The method won the first place in CVPR'21 competition, demonstrating its effectiveness.